# Differential Effect of Extracellular Acidic Environment on IL-1β Released from Human and Mouse Phagocytes

**DOI:** 10.3390/ijms21197229

**Published:** 2020-09-30

**Authors:** Petra Sušjan, Mojca Benčina, Iva Hafner-Bratkovič

**Affiliations:** 1Department of Synthetic Biology and Immunology, National Institute of Chemistry, Hajdrihova 19, 1000 Ljubljana, Slovenia; petra.susjan@ki.si (P.S.); mojca.bencina@ki.si (M.B.); 2EN-FIST Centre of Excellence, Trg Osvobodilne fronte 13, 1000 Ljubljana, Slovenia

**Keywords:** acidosis, IL-1β, inflammasome, species-specific, NLRP3

## Abstract

Areas of locally decreased pH are characteristic for many chronic inflammatory diseases such as atherosclerosis and rheumatoid arthritis, acute pathologies such as ischemia reperfusion, and tumor microenvironment. The data on the effects of extracellular acidic pH on inflammation are conflicting with respect to interleukin 1 beta (IL-1β) as one of the most potent proinflammatory cytokines. In this study, we used various mouse- and human-derived cells in order to identify potential species-specific differences in IL-1β secretion pattern in response to extracellular acidification. We found that a short incubation in mild acidic medium caused significant IL-1β release from human macrophages, however, the same effect was not observed in mouse macrophages. Rather, a marked IL-1β suppression was observed when mouse cells were stimulated with a combination of various inflammasome instigators and low pH. Upon activation of cells under acidic conditions, the cytosolic pH was reduced while metabolic activity and the expression of the main inflammasome proteins were not affected by low pH. We show that IL-1β secretion in mouse macrophages is reversible upon restoration of physiological pH. pH sensitivity of NLRP3, NLRC4 and AIM2 inflammasomes appeared to be conferred by the processes upstream of the apoptosis-associated speck-like protein containing a CARD (ASC) oligomerization and most likely contributed by the cell background rather than species-specific amino acid sequences of the sensor proteins.

## 1. Introduction

Local acidosis is the hallmark of many chronic inflammatory diseases. Dense infiltration of neutrophils and macrophages to the affected sites causes acidosis through massive amount of protons released through neutrophil respiratory burst and through an increased consumption of oxygen, which consequently results in hypoxia, a switch toward glycolytic metabolism, and accumulation of lactic acid [1]. A drop in local pH (pH 7–6) can be observed in pathologies such as ischemia [2] and atherosclerotic plaques [3]. Similarly, in patients with rheumatoid arthritis, the pH of synovial fluid is significantly lower than in healthy individuals [4]. Low extracellular pH and hypoxia are also characteristic features of the tumor microenvironment where accumulation of protons and development of hypoxic regions is enabled through high metabolic demand, hydration of CO_2_, and disorganized tumor vascular architecture [5]. The existing literature appears to be in dispute in terms of how pH affects the local inflammatory responses, as studies report both synergistic and antagonistic effects of acidosis on various inflammatory mediators. On one hand, slight acidosis was shown to enhance communication between complement pathways important for antimicrobial response to hospital-acquired infection with *Pseudomonas aeruginosa* [6]. Similarly, NO formation by murine macrophages was shown to be upregulated in slightly acidic conditions, mainly on a post-transcriptional level [7]. On the other hand, tumor necrosis factor (TNF) and interleukin 6 (IL-6) release from various murine cell lines was reported to be inhibited under acidic conditions [8]. Dispute may be attributed to the species-specific differences in innate immunity signaling. Extensive divergence between humans and mice was, for example, identified in liposaccharide (LPS)-regulated gene expression and attributed to its large dynamic range [9].

One of the main proinflammatory cytokines is IL-1β. Few studies have directly addressed its secretion pattern in human or mouse macrophages under acidic conditions, despite clear indications that a perturbation of ion homeostasis has a crucial role in the regulation of inflammasomes, particularly of NLRP3 (NLR family protein containing a pyrin domain 3) inflammasome [10,11,12] as one of the main molecular drivers of IL-1β secretion. Inflammasomes are multi-protein cytosolic platforms composed of sensor proteins, adaptor proteins ASC (apoptosis-associated speck-like proteins containing a caspase activation and recruitment domain, CARD) and precursor proteins procaspase-1, which rapidly assemble in response to diverse microbial molecules (PAMP) and damage-associated molecules (DAMP), often found at the site of chronic inflammation [13,14]. Autocatalytically activated caspase-1 in turn activates pyroptosis through gasdermin D cleavage and inflammatory cytokine IL-1β and IL-18 secretion from macrophages, thus eliciting a potent inflammatory response that recruits other immune cells to the affected site [15]. Extracellular acidosis was suggested to act as a novel NLRP3 inflammasome trigger upon observing a caspase-1 and potassium efflux-dependent IL-1β release from human monocytes and macrophages under mildly acidic conditions [16,17] as well as its inhibition under alkaline conditions [17]. In mice with colitis, on the other hand, IL-1β release was dampened when exposed to hypoxic conditions that often lead to acidosis. Some reports also suggest that acidosis differently modulates expression of IL-1β in monocytes and macrophages [16,18].

In this study, we aimed to systematically investigate the effect of the pathologically relevant pH decrease on the IL-1β secretion from various human- and mouse-derived monocytes/macrophages. We show that while mild decrease in pH alone caused substantial IL-1β secretion from human macrophages, mouse macrophages did not exhibit any IL-1β release. Rather, we observed a marked IL-1β secretion suppression when mouse macrophages were stimulated with various inflammasome instigators under acidic conditions. The inhibition was not a consequence of poor cell viability, protein degradation, or stunted protein expression/translation under acidic conditions. We show that pH sensitivity of IL-1β secretion in mouse macrophages occurs upstream of ASC speck formation, is reversible, and depends on the cell origin species. 

## 2. Results

### 2.1. Low Extracellular pH Stimulated IL-1β Maturation in Primed Human Cells But Not in Murine Macrophages

We first investigated the secretion pattern of IL-1β in human and mouse cell lines under various extracellular pH conditions. We started with human-derived BLaER1 B cells and transdifferentiated them into monocytes according to the published protocol [19]. BLaER1 monocytes were then primed with LPS and subsequently incubated in buffers with various physiologically and pathologically relevant pH (7.45, 6.5, 6, and 8). In accordance with the previous publication [17], cells responded to a decrease in pH from 6.5 to 6, with secretion of IL-1β already within 1 h of incubation (Figure 1a,b). As acidosis alone had at this time-point not yet stimulated maximal IL-1β secretion, further IL-1β increase was observed with the addition of potent triggers ATP and nigericin, regardless of pH. In the case of less potent particulate activators, acidosis triggered little to no additional IL-1β release (Figure 1b). Increase in IL-1β stimulation can only be observed in combination of particulate triggers with basic and neutral pH, the conditions that on their own do not stimulate substantial secretion of IL-1β (Figure 1b).

The same pattern of acidic pH-stimulated IL-1β release was observed in other tested cells of human origin, THP-1 cells, differentiated with phorbol 12-myristate 13-acetate (PMA) (Figure 1c) and peripheral blood mononuclear cells (PBMCs) (Figure 1d). Acidic medium was reported to trigger NLRP3 inflammasome [17]. Indeed, the addition of KCl, which counteracts the common NLRP3 inflammasome-activating process K^+^ efflux [10], inhibited IL-1β release in response to acidic conditions (Figure 1b). 

In contrast, when the same experiments were repeated on murine immortalized bone marrow-derived macrophages (iBMDMs), none of the pH conditions alone instigated any IL-1β maturation or pyroptosis within 1 hour (Figure 2a,b) or even after an extended period of incubation (Figure 3a). Cells appeared to be fully metabolically active after 1 h (Figure 2c) or 3 h (Appendix A), suggesting that low pH did not affect cell viability (Figure 2c). The lack of IL-1β release in response to extracellular acidification was observed also in primary mouse bone marrow-derived macrophages (BMDM) (Figure 2d) and immortalized murine microglial cells (Figure 2e). This suggests that species-specific differences may exist between murine and human cells in terms of their IL-1β inflammatory response to an acidic environment. 

To determine whether the cytosolic pH of the murine macrophages is changed under acidic conditions, we transduced mouse iBMDMs with a pH-sensitive green fluorescent protein (GFP) variant pHluorin. As this fluorescent protein has two excitation maxima depending on its protonation status (405 nm—deprotonated form, 488 nm—protonated form) [20,21], we were able to use the ratiometric cytometry method to measure the fluorescence in response to various pH conditions. When incubating the cells in acidic conditions alone, we observed a shift in the peak representing the fluorescence ratio between protonated and deprotonated forms of pHluorin towards the right, thus indicating a decrease in cytosolic pH (Appendix A). The ratio was shifted even further to the right when cells were treated with combined low pH conditions and various NLRP3 triggers, with the exception of the ionophore nigericin, which, as expected, equilibrated the extra- and intracellular pH through the pores (Appendix A). These results suggest the NLRP3 activators under acidic conditions further acidify the cell content of murine macrophages. The ability of murine macrophages to acidify the cytosol in response to acidic extracellular pH suggests that it is not the difference in response to pH that drives different effects on IL-1β secretion.

### 2.2. Under Acidic Conditions, Murine Macrophages Suppressed IL-1β Activation in Response to Various NLRP3 Inflammasome Triggers 

The inability of murine macrophages to secrete IL-1β in response to acidic pH prompted us to assess the effects of NLRP3 inflammasome triggers on the IL-1β activation under acidic conditions. Not only did murine iBMDM in the presence of low pH yield no IL-1β, its secretion and pyroptosis in response to nigericin were also markedly suppressed when stimulation was performed in the acidic buffer (Figure 2a,b). Furthermore, an increase in cell viability as judged from XTT assay was observed when nigericin treatment was performed in acidic pH, which correlated to a decrease in pyroptosis (Figure 2b,c). The same trend in IL-1β release suppression was observed with other murine phagocytes, primary mouse bone marrow-derived macrophages (BMDMs), and immortalized microglial cells (Figure 2d,e). The IL-1β maturation inhibition was further followed by Western blot (Figure 2f). While the amount of expressed cytokine precursor pro-IL-1β after incubation in acidic conditions alone was intact, thus excluding the effects on the precursor expression, we found a decreased amount of mature form of IL-1β (p17) in the cell supernatant as a result of nigericin treatment in acidic conditions. While very prominent with nigericin, this acidosis-related inhibitory effect was also observed with a wide array of other, chemically diverse NLRP3 instigators (Figure 3a) and was thus unlikely a consequence of potential trigger inactivation under acidic conditions.

### 2.3. IL-1β Activation in Response to Nigericin Was Rescued upon the Restoration of Physiological pH, Suggesting a Transient Effect That Does Not Depend on Irreversible Protein Modifications

The activation mechanism of many inflammasomes, particularly NLRP3, still remains elusive, and thus some key proinflammatory signaling mediators could be inactivated under acidic conditions. We therefore tested whether IL-1β activation can be rescued upon restoration of physiological pH. We first incubated iBMDMs with various pH buffers for 1 h. In the next hour, we replaced the buffers either with physiological pH buffer containing nigericin or with the same pH buffer containing nigericin. In negative control, we kept cells in various pH buffers for the duration of the experiment. As expected, the cells secreted significantly less IL-1β when activated with nigericin in the presence of low pH (Figure 3b). However, when stimulated with nigericin after low pH buffer was replaced with physiological pH buffer, cells preincubated in acidic pH and stimulated with nigericin in physiological pH secreted a comparable amount of IL-1β to the cells without previous exposure to low pH (Figure 3b). As the expression of the degraded signaling mediators upon restoration of the physiological pH could theoretically be renewed through residual LPS priming or IL-1β positive feedback loop, we repeated the reversibility experiment using a stable cell line, where NLRP3 expression was dependent only on doxycycline induction. A complete restoration of IL-1β secretion upon restoration of physiological pH was observed (Appendix A), further indicating that irreversible degradation due to acidic conditions is likely not the cause of decreased inflammasome activation. 

Intracellular acidification has been linked to the initiation of the apoptosis that is tightly associated with the proteasomal degradation of proteins [22]. Acidic conditions did not induce cell death in murine macrophages (Figure 2c and Appendix A). To fully eliminate the involvement of proteasomal degradation of proteins during acidic conditions, we treated iBMDMs with proteasome inhibitors before adding inflammasome triggers in the presence of low pH buffer. Proteasome inhibitors did not restore IL-1β release under acidic conditions (Appendix A), confirming that proteasomal degradation is not the reason behind the decreased inflammasome activation. Since the first activation signal (priming) of the inflammasome activation was performed under physiological pH conditions, it seems unlikely that protein translation could be affected by the pH change. While stimulation of cells with cycloheximide under physiological conditions resulted in a slight inhibitory effect on NLRP3 activation at physiological pH (Appendix A), the inhibition was not so profound to solely account for robust IL-1β inhibition observed with stimulation under acidic conditions. 

Inhibition of IL-1β in response to acidic conditions could potentially affect the opening or oligomerisation of sensor proteins through direct protonation of amino acid residues, particularly histidines, under acidic conditions. We reasoned that if the differential IL-1β activation pattern is specific to the amino acid sequence of NLRP3, the human variant of NLRP3 expressed on the mouse cell background will retain its IL-1β stimulatory phenotype in response to acidic conditions. However, when human NLRP3 was introduced into NLRP3-deficient mouse macrophages, it did not support IL-1β maturation upon acidosis, which suppressed IL-1β secretion in response to inflammasome instigators (Figure 3c). Further, we show that a previously identified shortest active truncation mutant lacking the LRR domain [23], mouse NLRP3^(1-686)^, and human NLRP3^(1-688)^ exhibited the same IL-1β inhibition pattern under acidic conditions as full length NLRP3 (Appendix A). Mutations in the gene encoding NLRP3 cause cryopyrin-associated periodic syndromes (CAPS). We show that acidic pH decreased constitutive cryopyrin-associated periodic syndromes mutation-associated IL-1β release, even when added after the inflammasome was already partially active (Figure 3d). Altogether, these data suggest that the differential pH sensitivity of IL-1β activation in response to acidic conditions is regulated by the cellular background, rather than being conferred by the sensor protein amino acid sequence.

### 2.4. Acidosis Inhibited IL-1β Release Due to NLRC4 and AIM2 Inflammasome Activation in Mouse Macrophages

In addition to the NLRP3, a palette of other cytosolic sensor proteins can initiate the assembly of inflammasomes and trigger IL-1β release. Among inflammasome sensors, NLRP3 is regarded as the sensor of cellular homeostasis, and thus the most likely component to be affected by acidic pH. Surprisingly, when cells were treated with Saty flagellin and poly(dA:dT), instigators of NLR family CARD domain-containing 4 (NLRC4) and absent in melanoma 2 (AIM2) inflammasomes, respectively, we observed similar acidosis-mediated IL-1β release inhibition (Figure 4a,b), suggesting that acidosis does not selectively affect NLRP3 inflammatory responses in murine cells. Some studies suggest that within the inflammasome complex, sensor proteins might coexist in heterocomplexes [24]. The apparent pH-dependent IL-1β inhibition in response to NLRC4 and AIM2 inflammasome activators could thus be caused by the inhibition of NLRP3 within the heterocomplex. To investigate this possibility, we treated NLRP3^−/−^ iBMDMs with NLRC4 and AIM2 triggers under various pH conditions but were in both cases still able to detect a significant acidosis-dependent decrease in IL-1β secretion (Figure 4c,d). This suggests that rather than being a NLRP3-specific occurrence, pH-sensitive processing of IL-1β is likely a more collective feature of the inflammasome signaling. Low IL-1β response was also not the consequence of ELISA detection malfunction under acidic conditions, as the IL-1β standard was normally detected even when combined with low pH stimulation buffer (Appendix A); moreover, the mouse IL-1β ELISA we used was not specific for the mature version of IL-1β, and thus pro-IL-1β and other species (e.g., p20) could also be detected. In addition, we verified our findings by replacing HEPES with bisTRIS propane buffer to ensure that the observed IL-1β and pyroptosis inhibition was not limited to the type of buffer used (data not shown). 

### 2.5. pH Sensitivity Occurred Upstream of ASC Speck Formation

We next focused on identifying the stage in the inflammasome activation cascade, affected by the acidic conditions. We hypothesized that the cytokine secretory mechanism might be dysfunctional, and thus we measured the upstream event—caspase-1 activation. As already expected, due to the acidosis-related inhibitory effect on the pyroptosis (Figure 2b), caspase-1 cleavage was also inhibited under acidic conditions (Figure 5a). As caspase-1 activation is upstream of cytokine maturation, the cytokine release is likely not alleviated directly due to the impaired secretory mechanisms. Next, we investigated the relationship between the pH and the adaptor protein ASC speck formation since it has been reported that in vitro formation of ASC filaments is favored by alkaline pH conditions whereas lowering the pH leads to their disassembly [25]. Indeed, visualization of immunofluorescently labelled endogenous ASC specks in fixed iBMDMs after they had been treated with nigericin in various pH buffers revealed that the formation of ASC specks was inhibited when stimulated under acidic conditions (Figure 5b). ASC protein expression was not affected with the incubation in low pH buffers alone, as shown by Western blot (Figure 5a). The expression of the sensor protein NLRP3 (Figure 5a) was similarly unaffected by acidosis. To confirm or refute the pH sensitivity at the level of the ASC speck formation, we activated NLRC4 inflammasome in the ASC-deficient iBMDMs. Namely, for the NLRC4 inflammasome activation, ASC is not obligatory as pro-caspase-1 can bind the activated NLRC4 sensor protein directly [26]. As the ASC-deficient cells responded in a similarly pH-sensitive fashion to activation under low pH (Figure 5c), we concluded that low pH-mediated defects in ASC oligomerization might originate from a process upstream of ASC, possibly at the level of the sensor protein.

## 3. Discussion

Extracellular pH is subject to change in various pathological conditions such as ischemia, hypoxia, atherosclerosis, and tumor microenvironment. The effect of acidosis on the prominent proinflammatory IL-1β cytokine secretion from macrophages is of specific interest as the main IL-1β regulators—inflammasomes—are implicated to play a contributing role in many of these conditions [27,28]. Conflicting literature reports on the relationship between the inflammatory responses and pH change might be due to the considerable species-specific differences in inflammatory response regulated by the innate immunity [9]. In this study, we systematically explored the effects of pathologically relevant acidosis on the IL-1β release in various human- and mouse-derived cell lines. 

We observed different IL-1β secretion patterns of human and mouse macrophages in response to changes in extracellular pH. While human macrophages exhibited an increase in IL-1β under mildly acidic pH that acted additively with NLRP3 triggers, which agrees with the previously reported findings [16,17], mouse macrophages secreted no detectable IL-1β in response to acidosis. Rather, low pH appeared to act antagonistically with various NLRP3 inflammasome triggers, as we observed markedly decreased IL-1β levels in response to soluble and particulate canonical triggers as well as in the case of CAPS-associated constitutive activation. A growing number of studies report human phagocyte-specific IL-1β secretory patterns that cannot be observed in mouse cells. LPS in human cells can stimulate potassium efflux and pyroptosis-independent IL-1β release in the absence of secondary trigger [19,29]. This has recently been reported to occur through a pathway involving Toll-like receptor 4 (TLR4), TIR domain-containing adaptor-inducing interferon-β (TRIF), receptor-interacting serine/threonine-protein kinase 1 (RIPK1), Fas-associated protein with death domain (FADD) and caspase-8 [19]. Furthermore, an additional level of NLRP3 inflammasome by alternative splicing was observed in human cells but not in murine cells [30].

We demonstrated that once the physiological pH conditions were reestablished, the activation of NLRP3 inflammasome was reversible, which indicates that such conditions did not irreversibly inactivate the key protein components of the inflammasome complex. Full reversibility of the process might also be suggestive of a protective negative regulation when cells are exposed to low pH in order to prevent inflammation-related damage. Acidosis is reported to have a protective role against cell death during ischemia reperfusion, as the fast normalization of pH paradoxically worsens the injury due to reactive oxygen species (ROS) formation and onset of the mitochondrial permeability transition pore [31]. Direct acidic infusion at the onset of reperfusion in a mouse model was reported to delay the recovery of normal pH and prevent the myocardium ischemia reperfusion injury [2]. Acidic extracellular conditions either alone (such as lactate) or in combination with NLRP3 activators were previously shown to lead to cathepsin D processing of pro-IL-1β yielding the p20 fragment [32,33,34]. As this fragment was proposed to be biologically less active [35], decreasing the pool of pro-IL-1β was suggested as a protective strategy. In our case, we observed the p20 fragment even at a neutral pH, suggesting that the major contribution to the decreased secreted levels of IL-1β originates from decreased p17 IL-1β. 

We showed that human NLRP3, when expressed in the mouse macrophages, did not support IL-1β activation under acidic conditions, but rather reflected the same IL-1β secretion phenotype found with mouse NLRP3. This suggests that differential response of cell species to acidosis is likely not specific to the amino acid sequence of the sensor protein. This fact is further supported by the observation that IL-1β release is also inhibited in response to stimulation of NLRP3^−/−^-immortalized bone marrow-derived macrophages with NLRC4 and AIM2 inflammasome activators under acidic conditions, which extends the pH sensitivity to NLRP3-independent IL-1β sources. 

Given the therapeutic implications of IL-1β and pyroptosis inhibition in the context of inflammasome-associated pathologies, we decided to identify the pH-sensitive stage of the inflammasome assembly cascade in mouse macrophages when exposed to extracellular acidosis. We found that caspase-1 activation and ASC speck formation were both inhibited while procaspase-1 and ASC protein levels were not affected, thus suggesting that an upstream stage of activation governs the pH sensitivity. The same phenotype was observed for ASC-independent, NLRC4-mediated IL-1β processing, thus suggesting that pH might affect sensor protein oligomerization. However, as ASC polymerization was pH-sensitive in vitro [25] and depended on ion fluxes in cell culture, where chloride efflux was shown to lead to dynamic inactive ASC aggregates [36], the effect of pH on ASC speck formation cannot simply be excluded. 

Our findings of differential inflammasome response to extracellular acidosis in human and mouse macrophages imply that caution is necessary when translating findings of mouse model research to the progression of the inflammatory pathologies in humans. Atherosclerosis mouse models where mice develop milder forms of the disease have already been shown to be of limited value unless transgenic mice are used [8]. This has been attributed to differential NF-κB pathway sensitivity to low pH [8]. Similar species-specific discrepancies occur in reports studying the effects of hypoxia, a condition accompanied with acidosis. While the mouse models of several chronic inflammatory conditions such as ulcerative colitis and Crohn’s disease provide mounting evidence that hypoxia resolves inflammation [28], studies on human donors or human cell lines conversely often find that hypoxic conditions increase inflammatory responses, e.g., by blocking the hydroxilation of factors within the NF-κB signaling pathway [37]. Similarly, exposure to conditions of low oxygen partial pressure associated with air travel or altitude-related hypoxia increase the risk of flare-ups in patients with irritable bowel syndromes (IBD) and found increased expression of inflammatory cytokines in the intestines of healthy individuals when exposed to high altitude-related hypoxia [38,39,40]. The results of our study emphasize that species-specific differences in pH sensitivity of IL-1β release should be considered when drawing conclusions from mouse models of chronic IL-1β-related inflammatory conditions. 

## 4. Materials and Methods 

### 4.1. Materials

The following chemicals and kits were used: cell culture media, fetal bovine serum (FBS), and other cell culture supplies (GIBCO, Thermofischer Scientific, Waltham, MA, USA); DMSO, ultra-pure LPS from Escherichia coli O111:B4, and Imject Alum (Thermofischer Scientific, Waltham, MA, USA); nanoSiO_2_ (nanoparticles of silica), naked poly (dA:dT), imiquimod and Pam_3_CSK_4_ (Invivogen, San Diego, CA, USA); nigericin, ATP, doxycycline, and XTT (Sigma, St. Louis, MO, USA); IL-1 beta mouse and human uncoated ELISA Kit (Thermofischer Scientific, Waltham, MA, US); Lipofectamine 2000 (Thermofischer Scientific, Waltham, MA, USA); paraformaldehyde (Electron Microscopy Sciences, Hatfield, PA, USA); SepMate isolation tubes (Stemcell Technologies, Vancouver, BC, Canada); Prolong Diamond Antifade solution with DAPI (Invitrogen, Thermofischer Scientific, Waltham, MA, USA); DOTAP Liposomal Transfection Reagent (Roche, Basel, Switzerland); Retro-X Tet-On 3G inducible expression system (Takara, Clontech, Kusatsu, Shiga Prefecture, Japan), LDH Cytotoxicity Assay (Roche, Basel, Switzerland). The following antibodies were used: purified anti-ASC antibodies (TMS-1, clone HASC-71) (Biolegend, San Diego, CA, USA), Alexa Fluor 633 goat anti-mouse IgG (H+L) (Invitrogen, Thermofischer Scientific, Waltham, MA, USA), anti-caspase-1 (p20) (mouse) Casper-1 (Adipogen, Liestal, Switzerland), HRP-conjugated goat anti-mouse IgG (H+L) polyclonals (Jackson ImmunoResearch, West Grove, PA, USA), rabbit polyclonal anti-mouse IL-1beta (GeneTEX, Irvine, CA, USA), goat polyclonal to rabbit IgG (HRP) (Abcam, Cambridge, UK), β-actin (8H10D10) anti-mouse mAb (Cell Signaling Technology, Danvers, MA, USA), and anti-NLRP3/NALP3 mAb Cryo-2 (Adipogen, Liestal, Switzerland).

### 4.2. Cell Cultures

Immortalized bone marrow-derived macrophages (iBMDMs) [41] and immortalized microglia [42] from C57BL/6 mice were a kind gift from Kate A. Fitzgerald and Douglas Golenbock (University of Massachusetts Medical School, Worcester, MA, USA), while BLaER1 [19,43] were a kind gift from Thomas Graf (Centre for Genomic Regulation, Barcelona, Spain). NLRP3^R260W^, human and mouse full length NLRP3, human NLRP3^1-688^, and mouse NLRP3^1-686^ (all previously reported by us [23,44]) were constructed on the NLRP3^−/−^ iBMDM background using a Retro-X Tet-On 3G inducible expression system (Takara, Clontech) as previously described [23]. THP-1 (from The European Collection of Authenticated Cell Cultures, ECACC, Wiltshire, UK), PBMCs, and primary mouse bone marrow-derived macrophages (BMDM) were cultured and seeded in RPMI medium supplemented with 10% FBS, while RPMI with 25 mM HEPES, 2% PenStrep, 1% L-glutamate (200 mM), 0.1% 2-β-mercaptoethanol (50 mM), and 10% FBS was used for BLaER1 cells. DMEM medium supplemented with 10% FBS was used for other cell lines. As previously reported, BLaER1 were transdifferentiated into monocytes using 100 nM β-estradiol E2 (Calbiochem, San Diego, CA, USA), 10 ng/mL hrIL-3 (PeproTech, Rocky Hill, NJ, USA), and 10 ng/mL hr-CSF-1 (M-CSF) for 5 days, while THP-1 were differentiated by an overnight incubation with 100 ng/mL phorbol 12-myristate 13-acetate (PMA) [19]. With the exception of priming (and differentiation) of PBMC, THP-1, BLaER1, and primary mouse BMDM that was performed in RPMI supplemented with 10% FBS, we performed all experiments in serum-free DMEM and stimulation buffer. 

### 4.3. Isolation of PBMC and Primary Mouse BMDM

PBMC were isolated from blood of healthy volunteers using SepMate isolation tubes according to the manufacturer´s instructions (Stemcell Technologies, Vancouver, BC, Canada) after obtaining an appropriate permit from the National Ethical Committee (0120-21/2020/4). Primary mouse BMDMs were isolated from femur and tibia bones of sacrificed mouse for which we obtained the postmortem tissue collection permit from the Administration for Food Safety, Veterinary Sector and Plant Protection (U34401-3/2018/4). Bone marrow was washed out of the bones with a PBS-loaded syringe and was centrifuged for 5 min at 1200 rpm. Erythrocytes were lysed by incubation in NH_4_Cl buffer for 15 min at 37 °C, followed by centrifugation and aspiration of the supernatant. The pellet was washed in PBS, and after centrifugation was resuspended in DMEM + 20% FBS. We plated 1 × 10^7^ cells per Petri dish and grew them in the presence of MCSF differentiation factor and PenStrep for 6 days. Cells were subsequently frozen. 

### 4.4. Inflammasome Stimulation

Cells were seeded in DMEM with 10% FBS at 1.6 × 10^5^ per well for the 96-well plate (ELISA, LDH, and XTT assays), at 1.6 × 10^6^ per well for the 6-well plate (Western blotting), and at 2.8 × 10^5^ cells per well for 8-well lides (immunofluorescence). The next day, cells were primed for 6 h with 100 ng/mL ultra-pure lipopolysaccharide (LPS) or overnight with 200 ng/mL Pam_3_CSK_4_. After medium removal, cells were stimulated with instigators in stimulation buffer of various pH values (8.1, 7.45, 6.5 and 6). Stimulation buffer was composed of 10 mM HEPES, 147 mM NaCl, 2mM KCl, 13 mM D-glucose, 2mM CaCl_2_x2H_2_O, and 1 mM MgCl_2_x6H_2_O, and was previously reported for use in inflammasome activation [45]. For K^+^ efflux inhibition, stimulation buffers were made with 134 mM KCl. Inflammasome instigators in stimulation buffer were added to the cells for 1 h (10 μM nigericin, 5 mM ATP) or 3 h (1 μg/mL poly (dA:dT)/Lipofectamine, 400 μg/mL alum, 180 μg/mL nanoSiO_2_, 3 μg/mL wild type (WT) Saty flagellin/DOTAP, 20 μM imiquimod, 1mM LLOMe). DOTAP and lipofectamine were used according to the manufacturer´s instructions. Saty flagellin was produced and used for stimulation as previously described [46]. As controls, we used DOTAP and lipofectamine without instigators. For experiments involving doxycycline-induced NLRP3 expression in the Retro-X Tet-On 3G inducible expression system (Takara, Kusatsu, Shiga Prefecture, Japan), we seeded cells in DMEM with 10% FBS at 1.2 × 10^5^ cells per well of a 96-well plate that was left overnight. Afterwards, NLRP3 expression and inflammasome priming were stimulated with a combination of 200 ng/mL Pam_3_CSK_4_ and 0.5 μg/mL doxycycline (DOX) for 9 h. Then, cells were stimulated for 1 h in buffers of various pH values alone or in the presence of 10 μM nigericin. Supernatants were collected and analyzed for IL-1β in ELISA assay.

### 4.5. Cytokine Detection Using ELISA Assays

IL-1β was detected in cell supernatants using mouse and human IL-1 beta uncoated ELISA Kit according to the manufacturer’s instructions (Thermofisher Scientific, Waltham, MA, USA). Multiplate reader SinergyMx (BioTek, Winooski, VT, USA) was used to measure absorbance.

### 4.6. Protein Detection Using Western Blotting

Cell stimulation was performed in a 6-well plate format. For caspase-1 and IL-1β detection, we concentrated the supernatant using 3K Amicon Ultra-0.5 mL centrifugal filters (Merck) and performed centrifugation at 14,000× *g* at 4 °C. Concentrate was mixed with SDS and β-mercaptoethanol loading buffer and was denatured at 90 °C for 10 min. For pro-caspase-1 and pro-IL-1β detection, we washed cells twice with cold PBS and lysed them. Protein concentration in the cell lysate was measured with BCA and mixed with SDS and β-mercaptoethanol loading buffer. A total of 30 μg proteins per well were separated on 12% or 15% SDS-PAGE gels, blotted onto the nitrocellulose membrane (GE Healthcare, Chicago, IL, USA), and detected using iBIND Automated Western System (Thermofisher Scientific, Waltham, MA, USA) with appropriate primary and secondary antibodies. For caspase-1 and pro-caspase-1 detection, we used the anti-caspase-1 p20 antibody Casper-1 (1:1000, Adipogen, San Diego, CA, USA), followed by HRP-conjugated goat anti-mouse IgG (H+L) polyclonal antibodies (1:600, Jackson ImmunoResearch, West Grove, PA, USA). IL-1β and pro-IL-1β were detected using rabbit polyclonal anti-mouse IL-1beta (1:1000, GeneTEX, Irvine, CA, USA) as a primary antibody and goat polyclonal to rabbit IgG (HRP) (1:600, Abcam, Cambridge, UK) as a secondary antibody. As a loading control reference in the case of lysates, we used either β-actin (8H10D10) mouse mAb (1:1000, Cell Signaling technology, Danvers, MA, USA) followed by HRP-conjugated goat anti-mouse IgG (H+L) polyclonals (1:600, Jackson ImmunoResearch, West Grove, PA, USA) or nonspecific band. SuperSignal West Femto or Pico Chemiluminescent Substrate (Thermo Scientific, Waltham, MA, USA) were used for detection of HRP-labeled bands.

### 4.7. Viability Evaluation Using XTT and LDH Assay

Freshly collected cell supernatants following stimulation were used for the end-point lactate dehydrogenase (LDH) activity measurement assay according to the manufacturer´s instructions (LDH Cytotoxicity Assay, Roche, Basel, Switzerland). On the cells, we performed XTT test to determine their viability/metabolic activity. For XTT assay, we used DMEM without phenol red and added to it a solution of tetrazolium salt (XTT) and phenazine methosulphate (PMS). Upon colour development we measured absorbance at 490 nm using a multiplate reader SinergyMx (BioTek, Winooski, VT, USA).

### 4.8. Confocal Microscopy

Immunofluorescent labeling of endogenous ASC was performed in fixed and permeabilized WT iBMDMs, as previously described [45,47]. ASC specks were detected using purified anti-ASC (TMS-1, clone HASC-71) primary antibodies (1:500, Biolegend, San Diego, CA, USA) and Alexa Fluor 633 goat anti-mouse IgG (H+L) (1:200, Invitrogen, Thermo Scientific, Waltham, MA, USA) as secondary antibodies. A Leica TCS SP5 laser scanning microscope mounted on a Leica DMI 6000 CS inverted microscope (Leica Microsystems, Wetzlar, Germany) with the HCX plan apo 63× (numerical aperture 1.4) oil immersion objective was used for imaging. A 405 nm laser line of 20 mW diode laser was used for DAPI excitation with emission between 415 and 450 nm. A 1 mW 631-nm HeNe laser was used for Alexa Fluor 633 anti-ASC excitation with emitted light detected between 640–660 nm. LAS AF (Leica, Wetzlar, Germany) and ImageJ software were used to acquire and process images.

### 4.9. Intracellular pH Estimation

In order to determine the change in cytosolic pH, we established a mouse macrophage cell line constitutively expressing a ratiometric green fluorescent protein (GFP) variant pHluorin that we previously optimized in our group [20,21], with dual excitation at 405 nm (deprotonated form) and 488 nm (protonated form). Cells were primed and stimulated with various triggers, upon which we measured the pHluorin fluorescence by ratiometric flow cytometry on the CyFlow Space cytometer (Partec, Germany). As described previously [48], we used split optics for 405- and 488-nm light path line. For excitation of protonated form, we used a blue solid-state laser that emits 50 mW of light at 488 nm and collected fluorescence using a standard 536/40-nm bandwidth band-pass filter (F488-nm channel). For excitation of the deprotonated form, we used a violet diode laser that emits 100 mW of light at 405 nm and detected fluorescence using a 480-nm dichroic mirror, with a 520/20-nm bandwidth band-pass filter (F405-nm channel) being used. A side-scatter signal was used as the trigger signal. We then gated the cells and analyzed at least 20,000 cells at the rate 200 cells/s.

## Figures and Tables

**Figure 1 ijms-21-07229-f001:**
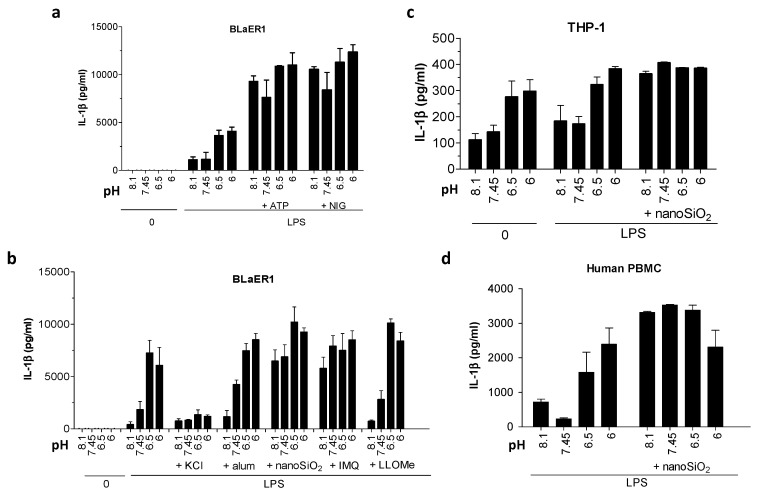
The effect of extracellular pH on interleukin 1 beta (IL-1β) release from human monocytes and macrophages. (**a**) IL-1β levels in supernatants after the differentiated BLaER1 monocytes were primed with 100 ng/mL lipopolysaccharide (LPS) for 6 h and then treated for 1 h with stimulation buffers of various pH values alone or containing 5 mM ATP or 10 μM nigericin. (**b**) IL-1β levels after BLaER1 were primed as in (**a**) and treated for 3 h with buffers with various pH values alone or containing 134 mM KCl, 400 μg/mL aluminum salts, 180 μg/mL nanoparticles of silica (nanoSiO_2_), 20 μM imiquimod or 1 mM L-leucyl-L-leucine methyl ester (LLoMe). (**c**,**d**) IL-1β levels by ELISA assay in supernatants after THP-1, differentiated with 100 ng/mL phorbol 12-myristate 13-acetate (PMA) or peripheral blood mononuclear cells (PBMCs) were primed with 100 ng/mL LPS for 6 h and exposed for 3 h to buffers with various pH values alone or containing 180 μg/mL nanoSiO_2_.

**Figure 2 ijms-21-07229-f002:**
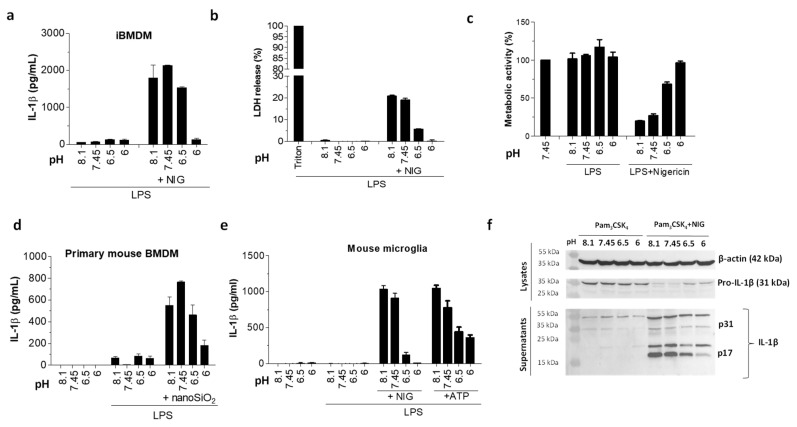
Decreased IL-1β maturation by acidosis in murine phagocytes. (**a**,**b**) IL-1β levels by ELISA assay (**a**) and lactate dehydrogenase (LDH) activity (**b**) in supernatants after immortalized bone marrow-derived macrophages (iBMDMs) were primed with 100 ng/mL LPS for 6 h and exposed to 10 μM nigericin in buffers with various pH values for 1 h. (**c**) Metabolic activity of cells from experiment in (**a**) was followed by XTT assay. (**d**,**e**) IL-1β levels by ELISA assay in supernatants of primary mouse bone marrow-derived macrophages (BMDMs) (**d**) and mouse microglia cells (**e**) after priming as in (**a**,**b**) and 3 h stimulation with 180 μg/mL nanoSiO_2_ or buffers alone (**d**), or 1 h stimulation with 10 μM nigericin, 5 mM ATP, or buffers alone (**e**). (**f**) Detection of IL-1β in supernatants and pro-IL-1β in lysates with Western blot after iBMDMs were treated overnight with 200 μg/mL Pam_3_CSK_4_ and 1 h with 10 μM nigericin or buffers alone.

**Figure 3 ijms-21-07229-f003:**
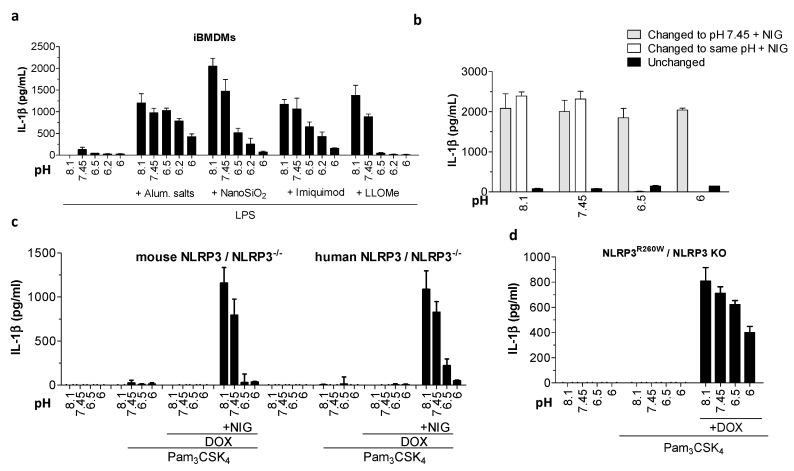
Acidic pH decreased IL-1β secretion due to NLRP3 inflammasome canonical and constitutive activation. (**a**) IL-1β levels by ELISA after iBMDMs were primed with 100 ng/mL LPS for 6 h and treated 3 h with buffers with various pH values alone or containing 400 μg/mL aluminum salts, 180 μg/mL nanoSiO_2_, 20 μM imiquimod, or 1 mM LLoMe. (**b**) IL-1β levels by ELISA assay in supernatants after iBMDMs were primed as in (**a**) and then incubated for 1 h in buffers with various pH values. Cells were then for another hour in either the same pH condition (unchanged), switched to a buffer with the same pH condition containing nigericin, or buffer with pH 7.45 containing 10 μM nigericin. (**c**,**d**) IL-1β levels by ELISA released from iBMDMs expressing reconstituted mouse or human NLRP3 (**c**) or NLRP3^R260D^ mutant (**d**) after overnight priming with 200 ng/mL Pam_3_CSK_4_ and 0.5 μg/mL doxycycline, followed by 1 h incubation in buffers with various pH values alone (**c**,**d**) or containing 10 μM nigericin (**c**).

**Figure 4 ijms-21-07229-f004:**
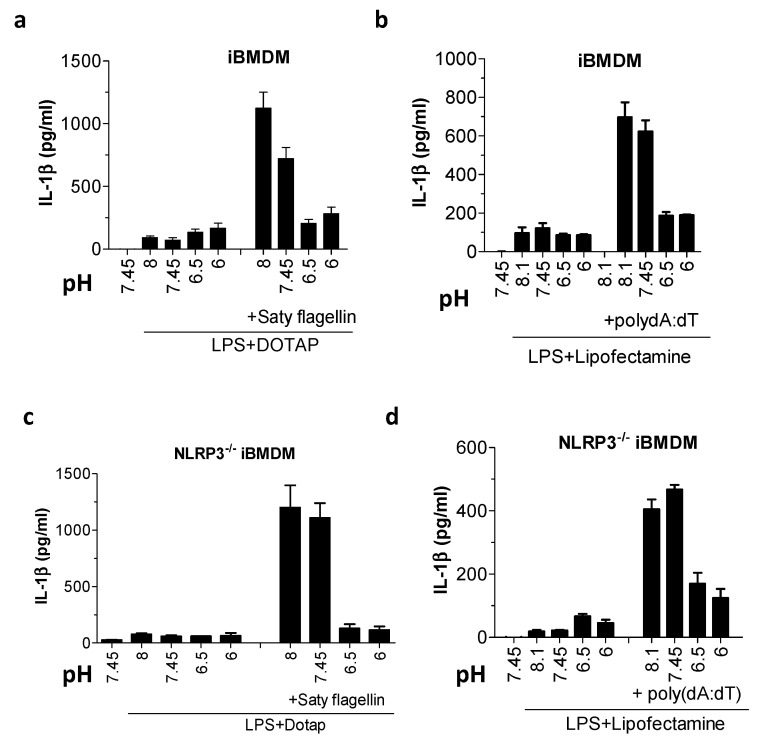
Acidic pH inhibited NLR family CARD domain-containing 4 (NLRC4) inflammasomes and absent in melanoma 2 (AIM2). (**a**–**d**) IL-1β levels by ELISA assay after wild type (WT) iBMDMs (**a**,**b**) or NLRP3^−/−^ iBMDMs (**c**,**d**) were primed with 100 ng/mL LPS for 6 h and then exposed to buffers with various pH values alone, containing 3 μg/mL Saty flagellin with DOTAP transfection reagent (**a**,**c**) or 2 μg/mL poly(dA:dT) with Lipofectamine transfection reagent (**b**,**d**) for 3 h.

**Figure 5 ijms-21-07229-f005:**
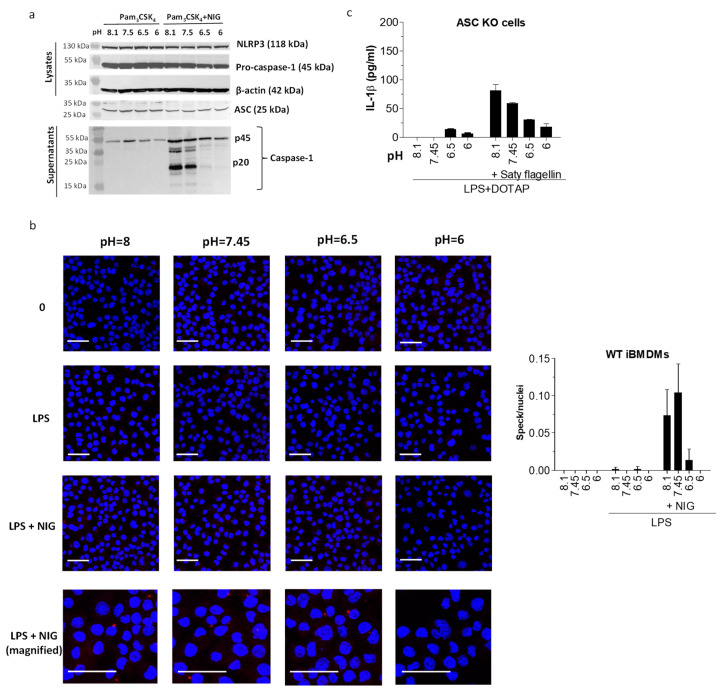
pH sensitivity was conferred upstream of apoptosis-associated speck-like protein containing a caspase activation and recruitment domain (ASC) oligomerization. (**a**) Detection of caspase-1 in supernatants and pro-caspase-1, NLRP3, and ASC in lysates with Western blot after iBMDMs were treated overnight with 200 μg/mL Pam3CSK4 and 1 h with 10 μM nigericin or buffers alone. (**b**) Detection of ASC specks (red aggregates) in immunofluorescently labelled WT iBMDM with confocal microscopy after cells were primed with 100 ng/mL LPS, followed by caspase-1 inhibition by 30 min treatment with 100 μM acYVAD and 1 h incubation in buffers with various pH values alone or containing 10 μM nigericin. Nuclei were stained with DAPI (blue). Scale = 40 μm. Images were quantified by determining the number of specks/number of nuclei ratio for 6 z-stack image frames for each experimental condition. (**c**) IL-1β levels by ELISA in supernatants of ASC^−/−^ iBMDMs after overnight priming with 100 ng/mL LPS and subsequent incubation in buffers with various pH values alone or containing Saty flagellin in DOTAP transfection reagent.

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
