# Peer review of "Differential Effect of Extracellular Acidic Environment on IL-1β Released from Human and Mouse Phagocytes"

_ijms, 2020, doi:10.3390/ijms21197229_

Round 1

Reviewer 1 Report

In this manuscript entitled "Differential effect of extracellular acidic environment 2 on IL-1β released from human and mouse phagocytes", authors investigated the effect of low pH on the inflammasome activation in murine phagocytes and human phagocytes.

Specifically, authors found that low pH severely dampened the inflammasome activation in murine phagocytes, while human monocytes/macrophages were unaffected by low pH.

Growing evidence support the differential activation/responses of the inflammasome between mouse macrophage and human monocytes/macrophages (Netea et al. Blood. 2009 Mar 5; 113(10): 2324–2335. Gaidt et al., 2016, Immunity 44, 833–846).

Differential regulation mechanisms by pH between mouse and human will have important implications, for various therapeutic approaches (e.g. inhibiting or activating NLRP3 in inflammatory diseases or cancers, characterized by the hypoxic/low pH environment).  Thus, the concept and findings in this manuscript are important.   

Experiments in this manuscript have been well-designed. Although key mechanisms remain to be fully elucidated, authors at least showed that low pH regulates the upstream of ASC-speck formation.  Additionally, authors carefully discussed results.  I would recommend for publication in the IJMS.

Reviewer 2 Report

This article by Sušjan et al., systematically investigated the differential effect of pathologically relevant pH decrease on the IL-1beta secretion from various human and mouse derived monocytes/macrophages. The authors highlighted the species-specific differences in IL-1beta secretion pattern in response to extracellular acidification. Finally, they show that IL-1beta secretion in mouse macrophages is reversible upon restoration of physiological pH. This pH sensitivity of NLRP3, NLRC4 and AIM2 inflammasomes can be regulated upstream of the ASC oligomerization and most likely contributed by the cell background. Overall, the conceptualization and flow of the story is logical but the experimental data is very weak and warrants further investigation. There are several caveats and misinterpretations that needs to be addressed in order to publish these findings.

  1. In Fig 1a, apparently no difference was observed in IL-1beta secretion, when cells were stimulated with LPS+ATP and LPS+NIG with varying pH, which suggests that there is no role of varying pH when inflammasome is activated. These IL-1b secretion data should be supplemented with western blot for Caspase-1 (pro and cleaved). In Fig. 1b why there is spontaneous IL-1b secretion without any stimulation? What is this antibody detecting? Cleaved or pro-IL-1b or both!!
  2. Since acidic pH was reported to trigger NLRP3 inflammasome activation, In Fig 2, why no activation was reported even after LPS+ATP and LPS+NIG stimulation? Are the cells dead by the time authors are measuring the cytokines? Normally these cells start dying 30 mins after ATP/NIG stimulation (which signifies inflammasome activation), authors are measuring after 1-3 hours post exposure. My guess is all the cells are dead and that could be the reason no IL-1b can be detected. This again need to be shown together with caspase-1 (pro and cleaved) blots.
  3. Authors mentioned that “The inhibition was not a consequence of poor cell viability, protein degradation or stunted protein translation or expression under acidic conditions”, whereas no experimental data was shown to support their claims. Intracellular acidic pH can also result in TRAIL-induced apoptosis and necroptosis. Thus, cell death and cell proliferation need to be shown (ex., LDH assay, Annexin/PI or by Incucyte).
  4. Check for the ASC oligomerization upon varying pH with and without inflammasome activation.
  5. In Fig 5b authors misinterpreted data and claim that “the formation of ASC specks was inhibited when stimulated under acidic conditions”, whereas no difference was observed in ASC spec formation. The cell number looks less which could be due to cell death (see earlier points) and the overall percentage of ASC spec is the same. Quantifying the spec data will help in visualization.

Round 2

Reviewer 2 Report

The manuscript has been substantially improved and all the major concerns were addressed. I feel the manuscript is now suitable for publication.